# From Incremental Test to Continuous Running at Fixed Lactate Thresholds: Individual Responses on %VO_2max_, %HR_max_, Lactate Accumulation, and RPE

**DOI:** 10.3390/sports11100198

**Published:** 2023-10-09

**Authors:** Daniel Fleckenstein, Jannik Seelhöfer, Nico Walter, Olaf Ueberschär

**Affiliations:** 1Department of Endurance Sports, Institute for Applied Training Science, Marschnerstraße 29, 04109 Leipzig, Germany; fleckenstein@iat.uni-leipzig.de (D.F.); seelhoefer@iat.uni-leipzig.de (J.S.); walter@iat.uni-leipzig.de (N.W.); 2Department of Biomechanics, Institute for Applied Training Science, Marschnerstraße 29, 04109 Leipzig, Germany; 3Department of Engineering and Industrial Design, Magdeburg-Stendal University of Applied Sciences, 39114 Magdeburg, Germany

**Keywords:** continuous running, training monitoring, training load, strain, lactate threshold running

## Abstract

With Norway’s successes in middle and long-distance running, lactate-guided threshold training has regained importance in recent years. Therefore, the aim of the present study was to investigate the individual responses on common monitoring parameters based on a lactate-guided conventional training method. In total, 15 trained runners (10 males, 5 females; 18.6 ± 3.3 years; VO_2max_: 59.3 ± 5.9 mL kg^−1^ min^−1^) completed a 40-min continuous running session at a fixed lactate threshold load of 2 mmol L^−1^. Lactate (La), oxygen uptake (VO_2_), heart rate (HR), and rating of perceived exertion (RPE) were recorded. The chosen workload led to lactate values of 2.85 ± 0.56 mmol L^−1^ (range: 1.90–3.80), a percentage of VO_2max_ utilization (%VO_2max_) of 79.2 ± 2.5% (range: 74.9–83.8), a percentage of HR_max_ utilization (%HR_max_) of 92.2 ± 2.5% (range: 88.1–95.3), and an RPE of 6.1 ± 1.9 (range: 3–10) at the end of the running session. Thereby, the individual responses differed considerably. These results indicate that a conventional continuous training method based on a fixed lactate threshold can lead to different individual responses, potentially resulting in various physiological impacts. Moreover, correlation analyses suggest that athletes with higher lactate threshold performance levels must choose their intensity in continuous training methods more conservatively (lower percentage intensity based on a fixed threshold) to avoid eliciting excessively strong metabolic responses.

## 1. Introduction

There are several physiological factors that are responsible for performance in middle and long distance running: the maximum oxygen uptake (VO_2max_) [1,2,3], the running speed at which VO_2max_ is reached (vVO_2max_) [2,4], running economy (RE) [5,6,7], a threshold capacity in terms of a lactate (LT) or ventilatory threshold (VT), or the speed at that threshold (vLT), respectively, refs. [8,9,10,11] as well as the utilization of VO_2max_ at a defined performance level (%VO_2max_) [12,13]. Current studies demonstrate a high level of consensus among international experts on a multitude of these factors, with additional parameters such as carbohydrate metabolism and glycolysis capacity also being mentioned [14]. To be able to develop these factors in a focused manner, training must be aligned with physiological training zones. In addition, monitoring is important in daily training in order to ensure compliance with these training zones. In principle, internal as well as external factors can be used for monitoring [15,16,17,18]. According to current research, there is no single parameter that can be regarded as a gold standard for monitoring [17,18,19]. Instead, it is a combination of internal and external factors that determines the training load. In addition, the so-called ‘fine-tuning approach’ is currently discussing how internal and external factors related to the conditions of the specific sports and events can be used more efficiently to further improve monitoring. The authors suggest that future studies should consider the best combination of monitoring tools for each specific sport setting [20]. Due to the practicability in training routine, lactate values (in addition to further parameters such as perceived exertion or heart rate (HR)) are increasingly used to determine training zones and are currently gaining in importance again [21]. It is common that the basis for lactate monitoring in training is the calculation of lactate thresholds, which are typically measured in laboratory settings. In addition, these laboratory data, or lactate values, respectively, are used in combination with zone models to define training zones and provide the basis for training intensities. Currently, this approach appears to be favored over other methods (e.g., using %VO_2max_ as the ‘reference’) [22].

Therefore, the aim of this study was to expose the participants to the same load, based on the velocity corresponding to a fixed lactate threshold value, in the context of a typical training method in running (40 min continuous running), and to determine how other monitoring parameters react to this training stimulus. Previous studies have already demonstrated that continuous running for 45–50 min at a fixed lactate threshold leads to premature termination of the load or ‘overly intense’ strain, resulting in a significant increase in lactate levels over time [23,24]. However, these fixed lactate thresholds were relatively high, reported as 4 mmol L^−1^ [23], or on the other hand, inconsistently with a range of 2–3 mmol L^−1^ [24]. Additional studies, although conducted on a cycle ergometer, revealed stable steady-state conditions at a fixed lactate threshold of 2 mmol L^−1^ [25], but with considerable individual variability that should not be underestimated. Some other physiological or subjective parameters, such as %VO_2max_ and perceived exertion, were not assessed. Accordingly, the aim of this study was to assess the individual physiological responses of trained athletes to a fixed threshold stimulus of 2 mmol L^−1^ during continuous running. In order to capture a holistic view of several common physiological markers and their interactions, the effects on VO_2_, HR, and rating of perceived exertion (RPE) were included in the evaluation. Based on the research and the increasing importance of elucidating individual adaptation responses [26], the following hypotheses are generated: (1) The same workload based on a fixed threshold (2 mmol L^−1^) leads to different levels of strain during continuous running among the participants; and (2) the measured parameters in the incremental test do not correspond to the measured values at the end of the continuous running.

## 2. Materials and Methods

### 2.1. Subjects

Fifteen trained runners (10 males, 5 females; Table 1) took part in this study. The inclusion criteria for this study were defined as follows: healthy male and female middle- and long-distance runners (national level), with an aerobic training load of at least 5 h per week and a VO_2max_ level exceeding 50 mL min^−1^ kg^−1^. Participants with acute viral or bacterial infections, existing orthopedic issues, or the use of pain medications were excluded. The participants were asked to avoid high-intensity and high-volume training sessions 48 h prior to the test days and were informed in detail about the study design. Moreover, this study was performed in accordance with the Declaration of Helsinki and was approved by the Ethics Committee of the Department of Engineering and Industrial Design at the Magdeburg-Stendal University of Applied Sciences under reference number EKIWID-2023-02-001FD.

### 2.2. Protocol and Test Design

The participants completed two test days in total. Figure 1 illustrates the experimental protocol design. On the first day, an incremental treadmill test was performed to determine the fixed lactate threshold of 2 mmol L^−1^. The initial running speed was determined based on individual performance levels (training results and race results over the last weeks). Additionally, all participants had completed at least one identical treadmill test, allowing the starting speed to be estimated in such a way that the first two stages were completed in the range below 2 mmol L^−1^, with the 2 mmol L^−1^ threshold being exceeded in one of the subsequent stages. To avoid unnecessarily prolonging the incremental test for the higher performing athletes, a uniform starting speed was omitted. The stage length was 2000 m, with four stages completed, increasing the speed by 0.25 m s^−1^ per stage. The rest period between stages was one minute. Subsequently, a second treadmill test, after 4 h of rest, was conducted to determine the maximum oxygen uptake (VO_2max_). Again, the starting speed was based on individual performance levels and based on previously completed VO_2max_-tests, with a stage duration of one minute and an increment of 0.15 m s^−1^ per stage until volitional exhaustion. In addition to volitional exhaustion, a leveling-off in VO_2_ was considered as a criterion for determining ‘a true’ VO_2max_. The goal was to achieve a cardiorespiratory exhaustion level between eight and ten minutes [27]. A 40-min continuous endurance run was completed three days (72 h) after the first test day. To avoid circadian or shoe-related effects [28], it was conducted at the same time of day (±30 min) and with the same running shoes. The running speed was determined based on the fixed lactate threshold of 2 mmol L^−1^ (vLa2) from the incremental treadmill test and kept constant during the whole time. In order to imitate the conditions of continuous running as a conventional training method, there was no break during the 40 min. All tests were performed on custom-built treadmills (POMA Maschinen- und Anlagenbau GmbH Poschendorf, Dürrröhrsdorf-Dittersbach, Germany) with 0% inclination. Before the tests, body weight and height were recorded (Seca Vogel and Halke Hamburg 910, seca GmbH and Co. KG, Hamburg, Germany). Breath-by-breath spiroergometry was used to measure respiratory gases using a stationary Metalyzer 3B system (CORTEX Biophysik GmbH, Leipzig, Germany). Before each test, during the rest periods in the incremental test, after the VO_2max_ test, and following the 40-min continuous endurance run, a 20 μL sample of arterial blood was taken from the earlobe, dissolved in 1000 μL of hemolysis solution, and analyzed to determine lactate accumulation (SUPER GL ambulance system; Dr. Müller Gerätebau GmbH, Freital, Germany). Heart rate (HR) was continuously measured using a chest strap (Wearlink W.I.N.D.; Polar, Kempele, Finland). The modified Borg CR10 scale was used to assess perceived exertion after each stage in the incremental treadmill test, after the VO_2max_ test, and after the continuous 40-min run [29].

### 2.3. Data Analysis

Spiroergometric data were processed using the MetaSoft^®^ 3.910 software (CORTEX Biophysik GmbH, Leipzig, Germany). VO_2max_ was defined as the highest measured value over 15 s [30]. For the incremental treadmill tests, an average for all spiroergometric data was taken over 60 s (within the last three minutes of the stage) for each stage to reduce potential measurement errors. Based on these values, a linear interpolation of VO_2_ was performed to obtain the VO_2_ values at vLa2. During the continuous 40-min test, the VO_2_ was similarly averaged over 60 s. In analogy to the respiratory data, the HR values in the incremental test and in the continuous test were also averaged over 60 s. In the incremental test, the data were linearly interpolated to determine a HR value at vLa2. The perceived exertion from the incremental test was linearly interpolated to determine a modified Borg CR10 value for vLa2. During the continuous test, the rating of perceived exertion was assessed at the end to determine RPE and to calculate session RPE. To determine vLa2, an exponential interpolation of the lactate curve was performed.

### 2.4. Statistics

A spearman correlation analysis was performed to examine relationships between the measured parameters from the incremental and continuous tests. Due to a small sample size and varying results for normal distribution (Shapiro-Wilk test), a non-parametric procedure was chosen. Scatter plots with a regression line including Spearman coefficient of correlation (r_s_), level of significance (*p*), and sample size (*n*) were used to visualize selected correlations. Owing to an assumed linearity, the Pearson correlation coefficient along with the level of significance, the coefficient of determination, and a regression line were also provided in the visualization. For all tests, the significance level was set at *p* < 0.05. Effect sizes ranging from 0.1 to 0.3 are considered as small, from 0.3 to 0.5 as moderate, and from >0.5 as large [31]. Data analyses were conducted with IBM SPSS Statistics 23 (IBM, Armonk, NY, USA). Data visualization was performed with R version 4.3.1 (The R Foundation, Boston, MA, USA). Results are presented as mean ± standard deviation. Additionally, minima and maxima are indicated.

## 3. Results

### 3.1. %VO_2max_, %HR_max_, and RPE at vLa2 during Incremental Test

In the incremental treadmill test, the utilization of maximum oxygen uptake (%VO_2max_) at vLa2 was on average 79.0 ± 2.1%, with an inter-individual range from 73.7 to 81.4%. The range of the utilization of the maximum heart rate (%HR_max_) at vLa2 varied from 84.1 to 91.4%, with a mean of 88.3 ± 2.2%. The rating of perceived exertion (RPE) at the fixed lactate level of 2 mmol L^−1^ was 4.5 ± 0.8. The minimum value was 3.4 and the maximum value was 5.9. To classify these recorded physiological factors of training control, a current zone model was introduced as a basis [32] (Figure 2). For %VO_2max_, it was observed that based on a three-zone model, the mean value is in Zone 1 (crossover area to Zone 2), with individuals ranging from Zone 1 to 2. The parameter %HR_max_, on average, falls within Zone 3, with individuals occupying Zone 2 or 3. The perceived exertion of the athletes, based on the model of Seiler and Kjerland [33], averaging at 4.5, falls within Zone 2, with individuals occupying Zone 1 and 2. The chosen fixed lactate parameter (2 mmol L^−1^) represents the “end” of Zone 1 or the “beginning” of Zone 2.

### 3.2. ∆%VO_2max_, ∆%HR_max_, ∆RPE, and ∆La from Incremental Test to Continous Running

The individual shifts in %VO_2max_, %HR_max_, RPE, and La of the participants—from the incremental test to the continuous running of 40 min—are presented in Figure 3. For ∆%VO_2max_, individual shifts range from −3.2 to +3.4%, with a mean deviation of 0.2%. Regarding the assignment of intensity zones, there are individual differences, with some athletes staying within their identified zones, while others show shifts from Zone 1 to Zone 2 and vice versa. Regarding ∆%HR_max_, the average shift for the whole group was +4.0%, with just one participant showing a slightly negative shift (−0.1%). The rest of the group exhibited significant increases, with a maximum of +8.1%. This resulted in all participants being in Zone 3 at the end of the 40-min continuous run.

A similar pattern was observed in terms of RPE. Only two participants reported lower perceived exertion after the 40-min run (−1.2 and −0.4). The rest of the group assessed the exertion as more intense at the end of the 40 min run as compared to the same load during the incremental test. On average, ∆RPE after the 40-min run was +1.6, with a maximal individual increase of +4.1. As a result, the majority of athletes shifted from Zone 1 to Zone 2 or even to Zone 3 and shifts from Zone 2 to 3 were evident. In terms of La values, constituting the fourth parameter of the training control presented, only one participant showed a decrease (−0.1 mmol L^−1^). The rest of the group had an average delta of +0.9 mmol L^−1^, with a maximal individual increase of +1.8 mmol L^−1^. Shifts in training zones were also evident in this parameter. Due to the fixed lactate level (2 mmol L^−1^), the participants had the same starting point, with all participants (except for one) ending up in Zone 2 at the end of the continuous run.

### 3.3. Correlations

The correlations among the physiological parameters from the incremental test are presented in Figure 4. A significant positive correlation with moderate effect size is observed between vLa2 in the incremental test and the lactate value at the end of the 40-min continuous run (La_T40_; *p* < 0.05, *R*^2^ = 0.37). Additionally, a positive statistical correlation between vLa2 and absolute oxygen uptake at vLa2 (VO_2abs, vLa2_) in the incremental test is evident, with a strong effect size (*p* < 0.001, *R*^2^ = 0.64).

Regarding the continuous 40-min run, correlations are presented in Figure 5. A significant relationship is observed between the lactate value at the end of the run (La_T40_) and the utilization of HR_max_ at the end of the run (%HR_max, T40_; *p* < 0.05, *R*^2^ = 0.39). There is no significant correlation between the parameters La_T40_ and the utilization of VO_2max_ at the end of the run (%VO_2max, T40_; *p* = 0.508). However, an association is found between La_T40_ and the rating of perceived exertion at the end of the run. RPE_T40_ is moderately correlated with La_T40_ (*p* < 0.05, *R*^2^ = 0.41).

## 4. Discussion

The results of the present study reveal that the individual patterns of strain among the runners differ significantly when applying a traditional training method (40 min of continuous running) aligned with a fixed threshold (2 mmol L^−1^). Such an approach leads to highly variable outcomes in the parameters of training monitoring, thereby altering the training effects substantially. This confirms our initial hypothesis (1), stating that the same workload leads to diverse strain levels among athletes. This finding is consistent with observations by Oyono-Enguelle et al. [25], who on a cycle ergometer also recorded an average lactate level of 2.1 mmol L^−1^ after a 45-min session at a fixed load of 2 mmol L^−1^, although with individual variations ranging from 1.44 to 3.12 mmol L^−1^. The research group of Föhrenbach et al. [24] found that in eight trained female marathon runners, after 45 min of continuous running at a fixed threshold load of 2.55 ± 0.35 mmol L^−1^, the final lactate values reached 3.5 ± 1.45 mmol L^−1^ (showing a ∆La from rest to min 45 of 2.01 mmol L^−1^), significantly exceeding the baseline value defined by an incremental test. In the present study, individual maximum values reached 3.80 mmol L^−1^, while the lowest value recorded was 1.90 mmol L^−1^.

Notably, however, other physiological parameters (%VO_2max_, %HR_max_, and RPE) responded very differently to the fixed workload. Therefore, the remaining hypothesis (2), that the measured parameters in the incremental test do not correspond to the measured values at the end of the continuous running, can be partially rejected and partially confirmed. On average, the difference in %VO_2max_ between incremental testing and the end of the 40-min of continuous running was merely 0.2% (range: −3.2 to +3.4%), whereas the difference in %HR_max_ was +4.0%. Perceived exertion was, on average, 1.6 scales higher after the 40-min run, and considerable higher values of 4.1 were recorded in individuals. This demonstrates that internal and external factors of training monitoring do not react identically to the presented load and, therefore, must be analyzed individually.

This becomes also evident when considering the fulfillment of training zones. While according to Haugen et al.’s model [32], at the end of the of the 40-min run, 13 out of 15 subjects are in Zone 2 (moderate-intensity training; MIT) based on lactate levels, considering the %VO_2max_ parameter, only 6 out of 15 subjects remain in Zone 2. Regarding the %HR_max_ parameter, no subject falls into Zone 2, as the values correspond to Zone 3 (high-intensity training; HIT). This illustrates the problem that training zone models provide a certain classification of parameters, but individual responses to stimuli may lead to an unclear assignment of a training zone in some cases (based on all parameters). This is mainly due to the fact that an incremental test (typically with short stage durations) cannot automatically be extrapolated to a classical training modality with significantly longer training durations.

There are certain influencing factors there that can affect parameters and provoke certain drifts. In the current literature, thermoregulatory capabilities, substrate metabolism, or muscle fiber type profile are mentioned as examples for such factors [34]. The authors currently refer to the concept of ‘durability’, defined as the ‘time of onset and magnitude of deterioration in physiological-profiling characteristics during prolonged exercise’ [34]. Even though ‘prolonged exercise’ usually refers to exercise durations in the range of hours, certain effects are evident even within the presented 40 min. To remain within a training zone, intensity or duration, or both, will need to be adjusted. This aspect should be considered for athletes and coaches when transferring incremental test designs and results to classical training methods such as continuous running.

In addition to individual responses to a fixed threshold stimulus, the presented correlations of the parameters also provide insights for training monitoring and mechanisms regarding training duration. The relationship between vLa2 in the incremental test and the lactate value at the end of the 40 min reveals a simple, yet important finding: the higher the recorded fixed lactate threshold in an incremental test, i.e., the running speed at 2 mmol L^−1^ lactate, the higher the lactate level at the end of continuous running. This means that the same fixed metabolic load in athletes with higher performance levels implies a higher shift, i.e., higher final lactate values, in a classical training modality. These data are consistent with previous observations [23]. Accordingly, these athletes should be controlled more cautiously during training, i.e., training should be conducted at lower intensities, to avoid leaving the desired training zones or provoking excessive metabolic loads. Furthermore, a strong correlation between vLa2 in the incremental test and absolute oxygen uptake (VO_2abs, vLa2_) is evident. This is logical, as the higher running speed entails increased energy expenditure, primarily covered by the aerobic system, i.e., oxygen uptake. This energetic output and substrate utilization must be considered in the context of training duration, especially during prolonged training sessions, and, as mentioned before, is currently discussed as a factor for shifts in physiological parameters in the context of ‘durability’ [35].

As mentioned earlier, it has been noted that the control of training through lactate levels is regaining importance [21]. The results of the present study demonstrate that a fixed workload at a defined lactate threshold leads to different responses on training monitoring, so that the parameters need to be individually considered and put into a ‘comprehensive picture’. However, there are also interrelations of parameters that can be useful for the training process. The lactate value at the end of the 40 min training session positively correlated with both %HR_max_ and RPE at the end of the training session. This means that higher metabolic strain is associated with higher cardiac output and perceived exertion. These relationships can be utilized in training in such a way that determining one parameter (e.g., HR) provides some further information about another parameter (e.g., La). Only for %VO_2max_, this was not the case, as no correlation with La_T40_ could be identified. From a practical perspective, the ‘fine-tuning approach’ by Boullosa et al. [20] can contribute to optimizing load management. The combination (‘triangulation’) of multiple parameters to the specific sport appears useful. It avoids relying solely on a single parameter to achieve training goals but rather views reaching the training zone as the result of the interaction of several suitable parameters. Furthermore, it seems important to assess the athletes’ so called ‘readiness’ before the training session to enhance training quality, a concept currently under discussion [36].

The following limitations of this study must be taken into account when interpreting the present results. The control of the 40-min training session was solely based on a fixed lactate threshold. While this approach was chosen intentionally, it did not consider the multidimensionality of other potential parameters (HR, VO_2_, RPE) as presented in the literature [26], suggesting that a one-sided focus on a single parameter may not allow for a comprehensive and adequate training monitoring. On the other hand, this also offers a perspective for further studies: how does multifactorial training intensity regulation (via several parameters) affect a typical training method in running? Secondly, the intensity of the continuous run was selected within an intermediate range, potentially falling between Zone 1 and Zone 2 according to zone models. Opting for a slightly more conservative intensity, i.e., slightly lower workload, might have resulted in more homogeneous outcomes across all subjects. Thirdly, although a conventional running duration of 40 min was implemented, much longer training durations are equally common, and effects on longer sessions will need to be investigated. Finally, the sample size of 15 subjects is within an acceptable range, but a larger number of athletes from middle and long distance running (national and international level) seems desirable in order to be able to make even clearer general statements about the results. In addition, we did not include the menstrual cycle of the female athletes, which could also have had an influence and should play a role in future investigations.

## 5. Conclusions

Although lactate-guided training is regaining importance, the present study shows that aligning the session monitoring with a fixed lactate threshold (based on an incremental treadmill test) in the area between low-intensity and moderate-intensity leads to individually varying strains during a 40-min continuous run. The La values vary from 1.9 to 3.8 mmol L^−1^, %VO_2max_ from 74.9 to 83.2%, %HR_max_ from 88.1 to 95.3%, and RPE from 3 to 10. On the one hand, this indicates that athletes’ responses are highly diverse, and relying solely on a fixed value will be insufficient. If the training goal is noticeably more extensive, i.e., in the low-intensity training zone, it is necessary to run with clearly lower speeds respectively with a markedly lower fixed lactate threshold. On the other hand, the shifts from incremental testing to a traditional prolonged training modality need to be considered and concepts integrating duration and intensity are needed [37]. This highlights the fact that the monitoring of daily training is dependent on individual responses, needs to take into account the duration of the workload, and requires adjustments in controlling based on a holistic parameter approach.

## Figures and Tables

**Figure 1 sports-11-00198-f001:**
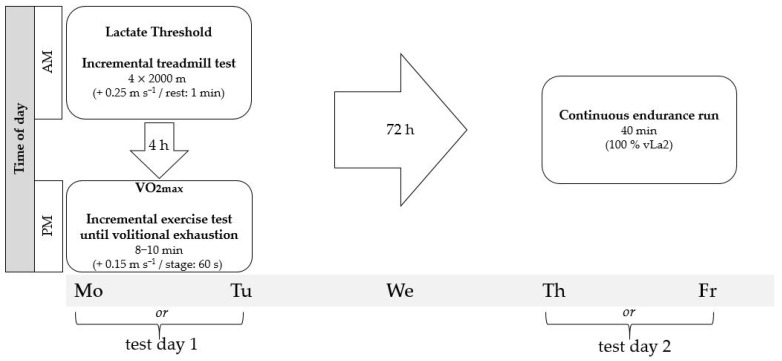
Experimental study design.

**Figure 2 sports-11-00198-f002:**
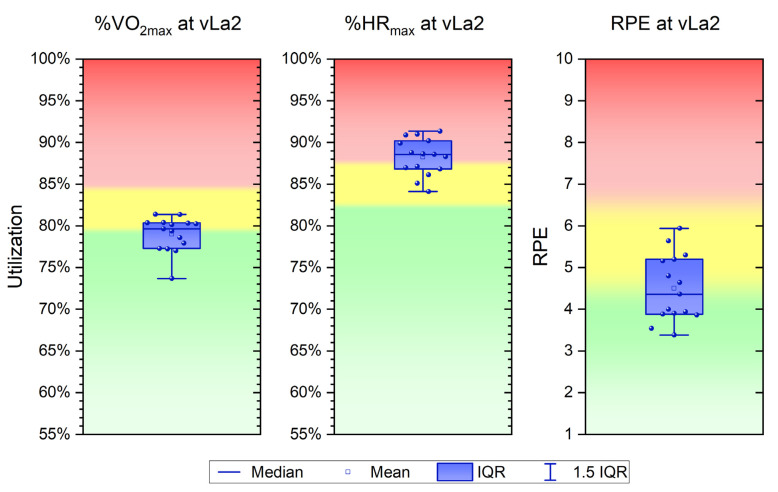
Utilization of maximum oxygen uptake (%VO_2max_), maximum heart rate (%HR_max_), and rating of perceived exertion (RPE) at the fixed lactate level of 2 mmol L^−1^ (vLa2) during incremental tests, on the basis of a current three-zone training model, where green = zone 1, yellow = zone 2, and red = zone 3.

**Figure 3 sports-11-00198-f003:**
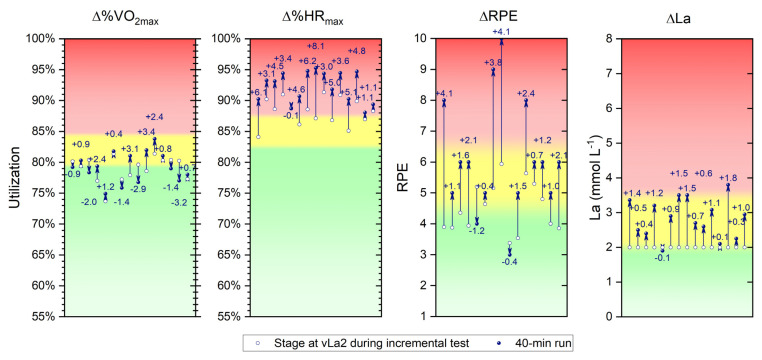
Individual shifts (∆) in the utilization of maximum oxygen uptake (%VO_2max_), maximum heart rate (%HR_max_), rating of perceived exertion (RPE), and lactate accumulation (La) from the incremental test to the continuous running of 40 min, on the basis of a current three-zone training model, where green = zone 1, yellow = zone 2, and red = zone 3.

**Figure 4 sports-11-00198-f004:**
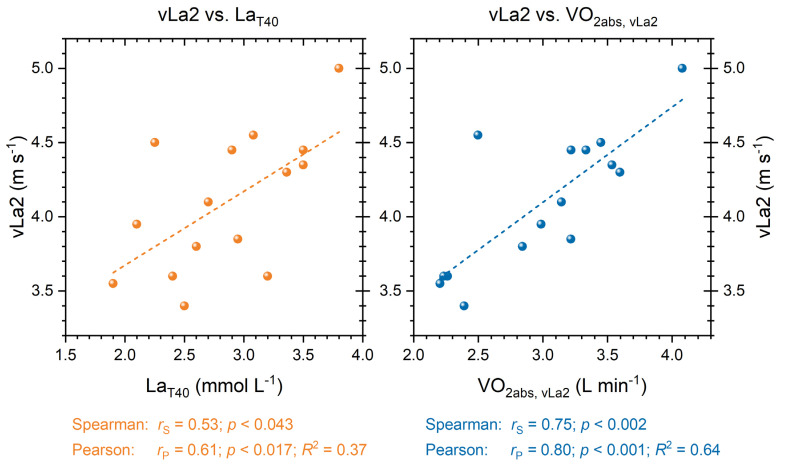
Correlations between fixed lactate level of 2 mmol L^−1^ (vLa2) during the incremental test and the lactate value at the end of the 40-min continuous run (La_T40_; (**left**)) as well as vLa2 and absolute oxygen uptake at vLa2 (VO_2abs, vLa2_) in the incremental test (**right**); regressions are presented in dashed lines.

**Figure 5 sports-11-00198-f005:**
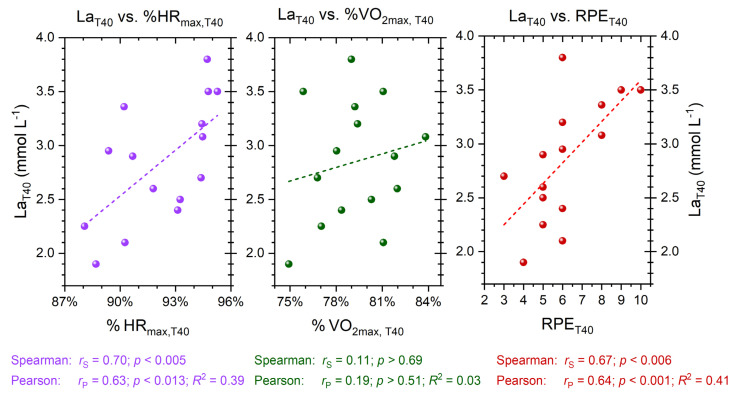
Correlations between the lactate value at the end of the 40-min continuous run (La_T40_) and the utilization of HR_max_ at the end of the 40-min continuous run (%HR_max, T40_; (**left**)), between La_T40_ and the utilization of VO_2max_ at the end of the 40-min continuous run (%VO_2max, T40_; (**middle**)), and between La_T40_ and the rating of perceived exertion at the end of the 40-min continuous run (RPE_T40_; (**right**)); regressions are presented in dashed lines.

**Table 1 sports-11-00198-t001:** General characteristics of participants included in this study.

Measure	All (*n* = 15) Mean ± SD (Min–Max)	Male (*n* = 10) Mean ± SD (Min–Max)	Female (*n* = 5) Mean ± SD (Min–Max)
Age (years)	18.6 ± 3.3 (15–25)	19.4 ± 3.4 (15–25)	17.0 ± 2.2 (15–21)
Body mass (kg)	63.6 ± 7.2 (50.2–72.9)	68.1 ± 2.8 (64.0–72.9)	54.7 ± 3.5 (50.2–59.5)
Body height (cm)	176.7 ± 8.2 (160–186)	181.5 ± 3.6 (175–186)	167.0 ± 4.8 (160–174)
VO_2max_ (mL min^−1^ kg^−1^)	59.3 ± 5.9 (50.6–69.6)	61.8 ± 4.4 (53.1–69.6)	54.3 ± 4.4 (50.6–59.8)
HR_max_ (beats min^−1^)	201 ± 10 (186–222)	201 ± 8 (186–212)	202 ± 11 (192–222)

## Data Availability

The data presented in this study are available on request from the corresponding author.

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
