# Peer review of "From Incremental Test to Continuous Running at Fixed Lactate Thresholds: Individual Responses on %VO2max, %HRmax, Lactate Accumulation, and RPE"

_sports, 2023, doi:10.3390/sports11100198_

Round 1

Reviewer 1 Report

This study aimed to investigate lactate-based training monitoring compared to conventional training methods. Fifteen trained runners participated in a 40-minute running session at a fixed lactate threshold load of 2 mmol L-1. Lactate (La), oxygen uptake (VO2), heart rate (HR), and rating of perceived exertion (RPE) were measured.

The results of this study showed that the chosen workload led to different lactate levels among the participants. These findings suggest that a conventional training method based on a fixed lactate threshold can result in varying individual responses, potentially leading to different physiological impacts.

Furthermore, correlation analyses suggest that athletes with higher lactate threshold performance levels need to choose the intensity in continuous training methods more cautiously (lower percentage intensity based on a fixed threshold) to avoid excessively strong metabolic responses.

This is an intriguing study that underscores the importance of individualizing training, particularly concerning the lactate threshold. It is worth noting that these results suggest there is no one-size-fits-all training method, and monitoring lactate levels can help tailor training intensity to an athlete's individual needs and capabilities.

There are several aspects that require further attention in this work:

1.            The Abstract can be improved by applying the following systematization: objectives, methods, results and conclusions.

2.            Introduction: Incorporate new references relevant to the research from the year 2022. Additionally, add a clearly defined goal for the study at the end of the introduction.

3.            Materials and Methods:

Explicitly describe the inclusion and exclusion criteria and clarify the method used to select the research group.

The small size of the research group raises concerns about the quality of the results and the ability to draw meaningful conclusions. Please consider addressing this concern.

Include a sample size calculation to justify the chosen group size.

Explain why representatives of both sexes were measured together without distinction.

4.            Discussion:

Ensure that the discussion aligns with the introduction and the reviewed literature. Incorporate more relevant references, especially from recent sources.

Focus the discussion on the main findings.

5.            Conclusion: The conclusion is where you should summarize your ideas and leave a strong final impression. Formulate clear and bold conclusions, dividing them into cognitive (despite limitations) and practical application.

Additional Suggestions:

Include the method of inclusion in the research group in the methods section.

Organize the entire study by graphically presenting the research methodology.

(line. 132,147) VO2max in subscript

These improvements will help enhance the clarity, rigor, and overall quality of the study.

Author Response

Response to Reviewer 1 Comments

Point 1: The Abstract can be improved by applying the following systematization: objectives, methods, results and conclusions.

Response 1: Thank you for the suggestion; we have clarified the study's objectives in the abstract. The methods, results, and conclusions are presented below.

Point 2: Introduction: Incorporate new references relevant to the research from the year 2022. Additionally, add a clearly defined goal for the study at the end of the introduction.

Response 2: With the studies from Casado et al. (2023) and Jones (2021), among others, we have included current literature. Nevertheless, we have used the advice to integrate a current study from the field of physiological profiling (Konopka et al., 2022) and monitoring (Boullosa et al., 2023). Furthermore, we have clearly described the aim of the study in the introduction (lines 69-73).

Point 3: Materials and Methods: Explicitly describe the inclusion and exclusion criteria and clarify the method used to select the research group.

Response 3: Thank you very much for this advice. We added a paragraph on the criteria for inclusion as well as exclusion of subjects (lines 81-85).

Point 4: The small size of the research group raises concerns about the quality of the results and the ability to draw meaningful conclusions. Please consider addressing this concern.

Response 4: See Response 5. Furthermore, we added a sentence in the limitation section.

Point 5: Include a sample size calculation to justify the chosen group size.

Response 5: Based on a similar study by Foxdall et al. (1996), which reported correlation coefficients of approximately r = 0.8 in the marathon group, a sample size of 9 would be obtained with an alpha error of 0.05 and a power of 0.8. However, since our study examined additional parameters, especially %VO2max, %HFmax, and RPE, and there were no similar studies available in that context, the sample size calculation was not performed. Assumed effect sizes of 0.6 to 0.7 would require sample sizes ranging from n = 13 to 19, which are within the range of this investigation. Nevertheless, due to the uncertainty surrounding these estimates, it was preferable to forego sample size calculation. (Furthermore, it should be noted that in the context of youth elite sports and our specific inclusion criteria, a larger sample size was not feasible under the given conditions.)

Point 6: Explain why representatives of both sexes were measured together without distinction.

Response 6: The focus of the current study was on the individual response of each athlete. The group statistics are included for completeness, but are not the major focus. Furthermore, the aim of the study was not to determine sex differences. In order not to further reduce the sample size, we decided to make the calculations and presentations for the entire sample. In addition, the data did not show any differences in terms of both sexes. A larger sample would have been desirable (in order to differentiate between the sexes if necessary), but this also proves difficult in competitive sport practice.

Point 7: Discussion: Ensure that the discussion aligns with the introduction and the reviewed literature. Incorporate more relevant references, especially from recent sources.

Response 7: We have expanded the discussion at various points and incorporated the literature added above, as well as additional recent studies.

Point 8: Focus the discussion on the main findings.

Response 8: As already laid out in Response 7, we did some additional parts in the discussion (highlighted in yellow).

Point 9: Conclusion: The conclusion is where you should summarize your ideas and leave a strong final impression. Formulate clear and bold conclusions, dividing them into cognitive (despite limitations) and practical application.

Response 9: We have supplemented the conclusion with an additional clear statement (lines 309-311) that leaves a distinct message. In our perspective, the points of individuality, training control and training duration (‘durability’) collectively make a clear statement (in the context of current research).

Point 10: Additional Suggestions: Include the method of inclusion in the research group in the methods section.

Response 10: We added a paragraph on the criteria for inclusion as well as exclusion of subjects (lines 81-85).

Point 11: Organize the entire study by graphically presenting the research methodology.

Response 11: To make the study design more easily comprehensible at first glance, we have included a figure (Figure 1) and added an additional sentence in the 'Protocol and Test Design' section.

Point 12: (line. 132,147) VO2max in subscript.

Response 12: Thank you for this advice. We have made this adjustment.

Reviewer 2 Report

I appreciate giving me the opportunity of reviewing interesting manuscript entitle : From incremental test to continuous running at fixed lactate thresholds: individual responses on %VO2max, %HRmax, lactate accumulation and RPE”. The following questions or suggestions are arises:

Abstract:

For indicating individual differences it is recommended that CI or range can be reported besides mean and SD.

Conclusion of abstract needs some modifications regarding the aims of the study.

Introduction

Regarding the sentences “However, these fixed lactate thresholds were relatively high, reported as 4 mmol L-1 [21], or inconsistently with a range of 2–3 mmol L-1” what was the significant or application of blood lactate with a range of 2-3 mmol L-1.

Protocols:

Table 1: first row please insert mean ±SD before min and max)

Line 77 to 78 the explanation of incremental treadmill test for determining intensity must be clarified or validity and reliability of the test must be clarified. This sentence must be explained” The initial running speed was determined based on individual performance levels.” How was initial performance of individuals determined?

Was menstrual phseses (follicular, luteal) of females considered in measurements?

The phrase of step length is not clear for me and needs to be explained.

What was the application of first and second incremental treadmill test?

How the lactate threshold was determined during the rest in incremental test and how it was observed during the training. Were there rest period during incremental test?

In exhausting incremental exercise test , a criteria for stopping the test is exhaustion. Do you think the Borg scale is applicable in this test or where did you apply that?

Results

Line 132 & 147 (titles): point after % seems must be deleted

I cannot see the number and explanations of figures .

Why both of spearman and Pearson correlation have been reported? What is the necessity.

Conclusion

Lines: 284-285 needs rewording

Author Response

Response to Reviewer 2 Comments

Point 1: Abstract: For indicating individual differences it is recommended that CI or range can be reported besides mean and SD.

Response 1: Thank you for the suggestion; we have added the range of values in the abstract.

Point 2: Conclusion of abstract needs some modifications regarding the aims of the study.

Response 2: Thank you for this note, which the other reviewer also included. We therefore described the clear goal of our study in the abstract right at the beginning.

Point 3: Introduction: Regarding the sentences “However, these fixed lactate thresholds were relatively high, reported as 4 mmol L-1 [21], or inconsistently with a range of 2–3 mmol L-1” what was the significant or application of blood lactate with a range of 2-3 mmol L-1.

Response 3: At this point we have only reviewed the work of the study group around Föhrenbach et al. (1987) described. So we can only speculate why this methodology was chosen. Since it was a very practical training setting, we assume that the research group was not allowed to give exact specifications or that the athletes were based on a pace and not on an exact physiological marker. This meant that only a certain rough range could be made possible and not a “comparable setting”.

Point 4: Protocols: Table 1: first row please insert mean ±SD before min and max).

Response 4: Mean ± SD was added.

Point 5: Line 77 to 78 the explanation of incremental treadmill test for determining intensity must be clarified or validity and reliability of the test must be clarified. This sentence must be explained” The initial running speed was determined based on individual performance levels.” How was initial performance of individuals determined?

Response 5: We have supplemented this description with some additional points (see lines 96-101). Based on previous tests, as well as training data and input from the coaches, we were able to determine the starting speeds individually. We deemed this necessary because the performance levels varied, and we wanted to avoid situations where some athletes would have to run for a very long time or where others would be quickly overwhelmed (which would have been the case with a uniform starting speed). It was important for us to capture the individual 2 mmol threshold of each athlete.

Point 6: Was menstrual phseses (follicular, luteal) of females considered in measurements?

Response 6: In the current study, the exact menstrual phase was not queried. The female athletes were only asked to report if they experienced any pain or discomfort. However, none of the participants reported such symptoms.

Point 7: The phrase of step length is not clear for me and needs to be explained.

Response 7: The term referred to 'stage duration' or 'stage length.' We have adjusted the term in the manuscript.

Point 8: What was the application of first and second incremental treadmill test?

Response 8: The aim of the first test was to determine the lactate thresholds, especially the 2 mmol/l lactate threshold, as this formed the basis for the 40-minute test. The second test was conducted to assess maximum oxygen consumption (VO2max) and maximum heart rate (HRmax), in order to have cardiorespiratory exhaustion parameters as markers for the exertion in the 40-minute test. To clarify this, we have provided a more detailed description in the text at some points (lines 95, 104-105).

Point 9: How the lactate threshold was determined during the rest in incremental test and how it was observed during the training. Were there rest period during incremental test?

Response 9: This is described in the 'Protocol and Test Design' as well as in the 'Data Analysis' sections. During the incremental test, a one-minute break was taken after each stage for lactate sampling. Exponential interpolation was performed on the values associated with each stage, and the exact point at 2 mmol/L lactate was determined by fitting the curve. During the 40-minute test, there were no breaks, and therefore, no intermediate lactate sampling.

Point 10: In exhausting incremental exercise test , a criteria for stopping the test is exhaustion. Do you think the Borg scale is applicable in this test or where did you apply that?

Response 10: That is absolutely correct, thank you. In addition to the volitional exhaustion, which represents one parameter for the VO2max test, the 'levelling-off' was also used as a conventional second parameter for determining a true VO2max. We have also added this to the manuscript (lines 107-108).

Point 11: Results: Line 132 & 147 (titles): point after % seems must be deleted

Response 11: Correct, thank you. We changed this.

Point 12: I cannot see the number and explanations of figures .

Response 12: That is also correct. We have added the captions to the figures.

Point 13: Why both of spearman and Pearson correlation have been reported? What is the necessity.

Response 13: We opted for this approach for the following reason: We calculated and included Spearman correlation values as usual. However, since the graph (the displayed regression line, which was crucial for our study's comprehension) suggests the use of a linear model, it makes sense to also provide the variance explanation R2, which inherently includes the Pearson correlation coefficient. Therefore, we included the Pearson correlation coefficient along with its alpha error. We hope this rationale is understandable. We have also correctly documented this in the 'Statistics' section for reader clarity.

Point 14: Conclusion: Lines: 284-285 needs rewording.

Response 14: Thank you, there was a missing word. We corrected this sentence.

Round 2

Reviewer 1 Report

I have no comments

Author Response

I have no comments

Response: Thank you for your positive feedback and the suggestions in the first review of this paper.

Reviewer 2 Report

Thanks for considering comments.

Two points are recommended in final revision/

1- I can not still see the number and caption of figures bellow them.

2-  Due to possible effect of menstrual phases (follicular, luteal) of females, its not controlling can be mentioned as a limitation of the study .

The manuscript needs minor revisions regarding quality of English language. 

Author Response

Point 1: I can not still see the number and caption of figures bellow them.

Response 1: Sorry for the inconvenience. The number and captions to the figures were sent to the editor, we thought this would also appear with you. So that you can definitely access them, attached are the Figure titles (and also supplementary in the newly uploaded paper):

Figure 1. Experimental study design.

Figure 2. Utilization of maximum oxygen uptake (%VO2max), maximum heart rate (%HRmax) and rating of perceived exertion (RPE) at the fixed lactate level of 2 mmol L-1 (vLa2) during incremental test, on basis of a current 3-zone-training model, where green = zone 1, yellow = zone 2 and red = zone 3.

Figure 3. Individual shifts (∆) in the utilization of maximum oxygen uptake (%VO2max), maximum heart rate (%HRmax), rating of perceived exertion (RPE), and lactate accumulation (La) from the incremental test to the continuous running of 40 min, on basis of a current 3-zone-training model, where green = zone 1, yellow = zone 2 and red = zone 3.

Figure 4. Correlations between fixed lactate level of 2 mmol L-1 (vLa2) during incremental test and the lactate value at the end of the 40-min continuous run (LaT40; left) as well as vLa2 and absolute oxygen uptake at vLa2 (VO2abs, vLa2) in the incremental test (right).

Figure 5. Correlations between the lactate value at the end of the 40-min continuous run (LaT40) and the utilization of HRmax at the end of the 40-min continuous run (%HRmax, T40; left), between LaT40 and the utilization of VO2max at the end of the 40-min continuous run (%VO2max, T40; middle) and between between LaT40 and the rating of perceived exertion at the end of the 40-min continuous run (RPET40; right).

Point 2: Due to possible effect of menstrual phases (follicular, luteal) of females, its not controlling can be mentioned as a limitation of the study.

Response 2: Thank you for pointing this out. We have included the potential limitation regarding the menstrual cycle of female athletes and added it to the paper (lines 328-330).
